# Non-small cell lung cancer microbiota characterization: Prevalence of enteric and potentially pathogenic bacteria in cancer tissues

**Nathan Dumont-Leblond**[1], **Marc Veillette**[1], **Christine Racine**[1], **Philippe Joubert**[1,2], **Caroline Duchaine**[1,3,4] *

**1** Centre de Recherche de l'institut Universitaire de Cardiologie et de Pneumologie de Québec, Quebec City (QC), Canada, **2** Département de Biologie Moléculaire, Biochimie Médicale et Pathologie, Université Laval, Quebec City (QC), Canada, **3** Département de Biochimie, de Microbiologie et de Bio-Informatique, Faculté des Sciences et de Génie, Université Laval, Quebec City (QC), Canada, **4** Canada Research Chair on Bioaerosols, Quebec City (QC), Canada

* Caroline.Duchaine@bcm.ulaval.ca

**Data Availability Statement:** The raw sequencing data in publicly accessible through the Sequence Read Archive (SRA) of the National Center for Biotechnology information (NCBI) with the

## Abstract

Following recent findings linking the human gut microbiota to gastrointestinal cancer and its treatment, the plausible relationship between lung microbiota and pulmonary cancer is explored. This study aims at characterizing the intratumoral and adjacent healthy tissue microbiota by applying a 16S rRNA gene amplicon sequencing protocol to tissue samples of 29 non-small cancer patients. Emphasis was put on contaminant management and a comprehensive comparison of bacterial composition between cancerous and healthy adjacent tissues of lung adenocarcinoma and squamous cell carcinoma is provided. A variable degree of similarity between the two tissues of a same patient was observed. Each patient seems to possess its own bacterial signature. The two types of cancer tissue do not have a distinct bacterial profile that is shared by every patient. In addition, enteric, potentially pathogenic and pro-inflammatory bacteria were more frequently found in cancer than healthy tissue. This work brings insights into the dynamic of bacterial communities in lung cancer and provides prospective data for more targeted studies.

## Introduction

The pulmonary microbiota is altered in many cases of lung pathologies such as in COPD (Chronic Obstructive Pulmonary Disease) [1,2], idiopathic pulmonary fibrosis and cystic fibrosis [3]. The lung cancer tissue is also colonized by specific bacterial communities [4,5]. Composition of the local microbiota have a significant impact on the initiation, the development, the migration, and the treatment of different types of cancer, including colorectal [6,7] and pancreatic cancer [8–10]. Such observations on the influence of the lung microbiota in pulmonary cancer as yet to be confirmed but are plausible. Bacteria in the digestive tract can induce a chronic inflammatory state that is favorable to cancerogenesis and cancer growth by

accession number: PRJNA680529. The S1 Table details the accession numbers of each sample. Underlying data of the figures of the article, as well as the OTU and taxonomic tables, are made available in the Supporting information file.

**Funding:** The Fonds sur les maladies respiratoires J.-D.-Bégin-P.H.-Lavoie and the Quebec Respiratory Health Network (QRHN) funded this work (PJ and CD). Two different master's degree scholarship, from the Engineering Research Council of Canada (NSERC) and the Fonds de Recherche du Quebec – Santé (FRQS), were awarded to NDL. He also received short internship scholarships from the NSERC and QRHN. The funders had no role in study design, data collection and analysis, decision to publish, or preparation of the manuscript.

**Competing interests:** The authors have declared that no competing interests exist.

the production of toxins, reactive oxygen species (ROS) or the modulation of human metabolic pathways [11–14]. The direct production of genotoxic toxins by bacteria can also induce cancer [15,16]. The microbiota composition stability observed between paired primary and metastatic tumors also suggest a role in cancer migration. In renal cell carcinoma, small cell lung cancer and melanoma, the microbiota can alter the response to immunotherapy via antibodies targeting programmed cell death protein-1 (PD-1) or its ligand (PDL-1) [17–19]. Bacteria are also believed to participate in chemoresistance through the modulation of autophagy and the enzymatic degradation of the antineoplastic drugs such as gemcitabine [7,20]. Therefore, the pulmonary microbiota can potentially influence the carcinogenesis and treatment of lung cancer [21].

This study aimed at deploying a newly validated methodology to identify the bacterial components of the lung microbiota in cancer patients [22]. Tumoral and healthy tissue samples obtained from patients diagnosed with squamous cell carcinoma or adenocarcinoma that underwent lung resection were characterized. The bacterial signature of cancerous and healthy tissue of a same patient (intra-patient) or different patients (inter-patient) were compared. The taxonomic identification of bacteria frequently found in cancerous and healthy tissue were also contrasted. Enteric and potentially pathogen genera were found more frequently in cancerous tissues. Understanding the composition of the human lung microbiota and the way it is shaped will provide insights into the relationship between bacteria and the development of lung cancer. Such knowledge could further our ability to establish appropriate preventive measures and correctly treat cancer patients, therefore improving their prognosis.

## Materials and methods

### Patient enrollment and sampling

Twenty-nine patients that underwent a lobectomy for pulmonary squamous cell carcinoma or adenocarcinoma were enrolled by written consent under the biobank approbation framework of the Institut Universitaire de Cardiologie et de Pneumologie de Quebec (IUCPQ) (number 1200). This study was approved by the IUCPQ ethic committee and regulations were followed. The patients needed to be free of antibiotics or neoadjuvant therapy 3 months prior the surgery and have a tumor larger than 2 cm in diameter. Clinical data are presented in Table 1.

### Samples treatment

The sampling, sample treatment, bacterial DNA extraction, sequencing and bioinformatics analyses were performed as we described Dumont-Leblond *et al.* [22]. Briefly, a whole section of tumors of 5 mm thickness, 8 cm³ of healthy tissue located 5 cm from tumors in the same pulmonary region, and a methodological control accounting for every step of the protocol were extracted and sequenced for each patient. The samples were enzymatically and mechanically homogenized using the Liberase™ TM enzyme cocktail (Roche, Bâle, Switzerland) and the Fisherbrand™ 150 homogenizer with plastic probes (Thermo Fisher Scientific, Waltham, USA). Then, bacterial DNA was extracted using an adapted protocol of the QIAamp® DNA Blood Maxi Kit (QIAGEN, Hilden, Germany). The Illumina MiSeq platform was used to sequenced V3-V4 16S amplicons prepared in a dual-indexed fashion with the primers Bakt_805R-long and Bakt_341F-long described by Klindworth *et al.* [23].

### Contaminants management

The protocol used encompasses many features to ensure the detected microbiota would not be modified by contaminants incorporated experimentally. The risks of contamination were

**Table 1. Patient's clinical data.**

| Identification number (ID) | Age | Sex | Smoking status** | Lobe/Localization* | Tumor size, width/length (mm) | Histologic type* | Pathological stage |
|---|---|---|---|---|---|---|---|
| 1 | 80 | F | Ex-smoker (2015, 31) | RUL | 31/22 | AC | 1B |
| 2 | 62 | F | Non-smoker | RLL | 45/40 | SqCC | 2A |
| 3 | 77 | F | Non-smoker | LLL | 25/20 | AC | 1A3 |
| 4 | 70 | M | Ex-smoker (2016, 58) | LLL | 28/25 | SqCC | 1B |
| 5 | 78 | M | Ex-smoker (1984, 54) | RLL/RML | 70/70 | SqCC | 3B |
| 6 | 63 | F | Ex-smoker (2013, 22.5) | RUL | 32/28 | SqCC | 1B |
| 7 | 64 | F | Ex-smoker (2000, 3.3) | RLL | 28/24 | SqCC | 1B |
| 8 | 62 | M | Ex-smoker (2019, ND) | RML | 36/25 | SqCC | 1B |
| 9 | 73 | M | Ex-smoker (1975, 51) | RUL | 35/35 | SqCC | 1B |
| 10 | 77 | M | Ex-smoker (1973, 19) | RUL | 35/28 | AC | 1B |
| 11 | 54 | M | Ex-smoker (2019, 86) | RML | 38/28 | AC | 3A |
| 12 | 65 | F | Ex-smoker (1983, 52.5) | LUL | 23/21 | AC | 1A3 |
| 13 | 58 | M | Active smoker (33.75) | LUL | 26/24 | SqCC | 1A3 |
| 14 | 66 | F | Ex-smoker (2018, ND) | RUL | 30/17 | AC | 1A3 |
| 15 | 65 | F | Ex-smoker (2019, 51) | RLL | 75/35 | AC | 3A |
| 16 | 80 | F | Ex-smoker (2018, 32.5) | RUL | 32/22 | SqCC | 1B |
| 17 | 67 | M | Active smoker (50) | RUL | 45/35 | AC | 2B |
| 18 | 62 | F | Ex-smoker (1975, 5) | RUL | 35/25 | AC | 3A |
| 19 | 74 | M | Ex-smoker (2018, 56) | LUL | 44/40 | AC | 2B |
| 20 | 66 | M | Active smoker (106) | LUL | 21/21 | SqCC | 1A3 |
| 21 | 66 | M | Ex-smoker (1996, 30) | LLL | 37/37 | AC | 1B |
| 22 | 68 | F | Ex-smoker (2013, 57.5) | RUL | 20/20 | SqCC | 1A2 |
| 23 | 50 | M | Active smoker (38) | RUL | 56/45 | AC | 2B |
| 24 | 70 | F | Active smoker (8.1) | RUL | 27/27 | AC | 1B |
| 25 | 58 | F | Ex-smoker (2019, 44) | RUL | 43/32 | AC | 1A2 |
| 26 | 73 | F | Ex-smoker (1999,18) | LLL | 23/20 | AC | 3A |
| 27 | 70 | F | Ex-smoker (2015, 50) | RLL | 63/63 | AC | 2B |
| 28 | 70 | F | Ex-smoker (2003, 35) | LUL | 32/32 | AC | 1A3 |
| 29 | 71 | F | Ex-smoker (1986, 23) | LUL | 32/22 | AC | 2B |

*F = female, M = male, A = adenocarcinoma, SqCC = squamous cell carcinoma, LLL = left lower lobe, RLL = right lower lobe, RUL = right upper lobe, RML = right middle lobe.

** Year they quit smoking and/or number cigarette pack-year (estimation of the total number of packs smoked in one's life).

minimized during sampling by selecting only laparoscopic lobectomy procedures, by optimizing the organ transportation, by using sterile sampling equipment only and assuring quick sampling and storage. A single experimental control was implemented for each patient that accounts for every step of the experimental protocol, from sampling to sequencing. The DNA extraction method is also designed to be completed in a single tube and by the addition of reagent, which allows us to obtain representative no-template control for each pair on healthy and cancerous tissue. The presence of contaminants was accounted for through bioinformatics by removing Operational Taxonomic Unit (OTU) found in controls from the corresponding samples on a relative abundance basis [22,24].

## Sequences processing

The sequencing data was cleaned and clustered into OTUs following Mothur SOP version 1.40.5 with the SILVA 16s rRNA gene database release 132 [25–30]. A custom contaminants

removal method [24] was applied before or after diversity and differential abundance analyses were performed in RStudio [31]. Plots were created using ggplot2 and ggpubr, versions 3.2.0 and 0.2.1 [32,33].

## Statistics and reproducibility

Alpha and beta diversity (Bray-Curtis distances on relative abundances) computations were performed using the Vegan package, version 2.5–5, as well as analyses of variance (ADONIS) at 5000 permutations [34]. Stratification or pairing in the statistical tests were performed when necessary accounting for the patient variable. Differential abundance tests for microbial abundance at OTU levels were done using the DESeq2 package [35] with modified models for individuals nested within groups as described by Love *et al.* when needed [36]. OTUs not identified at the specific taxonomic level were removed and the rest was agglomerated by name. OTUs or taxonomic units with fewer than 4 reads in at least 3 samples and with a variance across the whole data set lower than $10^{-4}$ were removed to limit the amount of abundance test ran and statistical correction applied to the output. Pearson's correlations were performed on relative OTU abundances between tissue samples from a same patient using the metagenomeSeq package version 1.24.1 [37]. The relative abundance values used were zero-inflated and not normally distributed ($p < 0.05$, Shapiro-Wilk). In the absence of appropriate alternative, this test has still been proven serviceable in these conditions [38]. The significance threshold of every statistical test was set at 0.05. Venn diagrams were obtained using ampvis2 package, version 2.5.8 [39]. Only OTUs with abundance of at least 0.001% were considered and singletons were removed. Only OTUs found in a minimum of 30% of samples in each category were considered as "Core".

## Results

Twenty-nine patients with non-small cell lung cancer undergoing lobectomy were recruited (Table 1). For each patient, the resected pulmonary lobe was subsamples to collect tumoral tissue and adjacent healthy tissue. A single negative control, accounting for every step of the protocol, was included with each pair of tissues.

### Contamination and controls

The bacterial profiles of the tissue samples were compared to the corresponding methodological controls in order to assess the influence of contaminants on the lung microbiota detected. The Principal Coordinates Analysis (PCoA) computed from weighted (presence and proportion of OTUs) and unweighted (only absence or presence of OTUs) Bray-Curtis distances displayed in Fig 1 shows great disparity in bacterial profiles between tissue samples and controls. This observation is confirmed by a highly significant analysis of variance tests performed stratified by patients on weighted ($p = 0.0002$, $R^2 = 0.11441$) or unweighted ($p = 0.0002$, $R^2 = 0.03486$) distances. Therefore, samples and their control share very little OTUs that may also be present in different proportions.

### Patient and clinical variable influence on the bacterial profiles

Bacterial profiles (β diversity) were identified, for tissues coming from a same patient (intra-patient), to be more similar to those of different patients (inter-patient). A PCoA from unweighted Bray-Curtis distances reveals clustering of the samples by patients (Fig 2), which is confirmed by a variance analysis test (ADONIS) with p-value of 0.0002 and $R^2$ of 0.03332. Therefore, this correlation has extremely low chances of being fortuitous, but only accounts

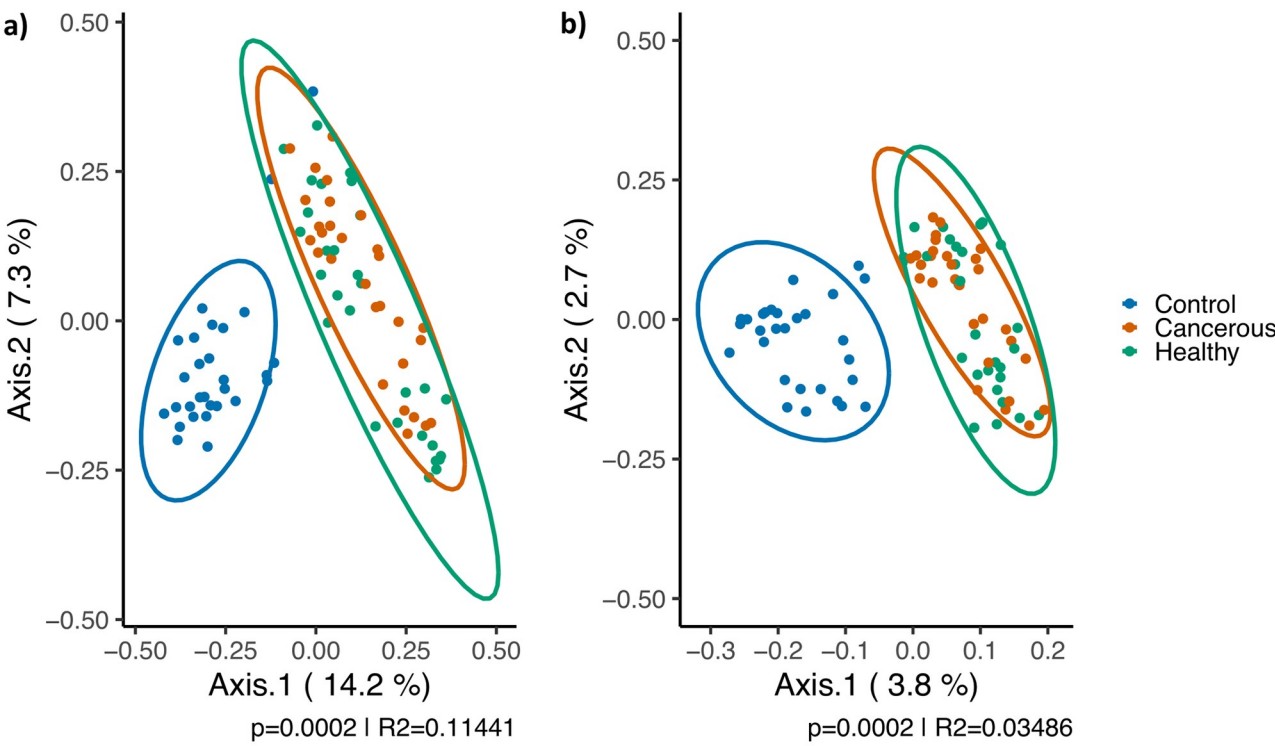

**Fig 1. Principal coordinates analysis of the extraction controls compared to the samples before the removal of contaminant OTUs. A**. The positions are based on weighted Bray-Curtis distances **B**. The positions are based on unweighted Bray-Curtis distances. An analysis of variance test (ADONIS) stratified by patients was performed between the tissues and control to obtain the p-values and $R^2$ displayed. Each dot represents a control or tissue sample. Controls are identified in blue and cancerous and healthy tissues in orange and green, respectively. Multivariate t-distribution at 95% was used to compute the data ellipses. n = 29.

for a very small part of the overall variation (3.332%). In an attempt to explain the residual variation observed in Fig 2, the different clinical variables correlated to the microbial compositions by a similar variance analysis. The type of tissue (cancerous vs. healthy) dependently of the type of cancer did not significantly explain additional variation in bacterial profiles when stratified by patients (adenocarcinoma: p = 0.42352, $R^2$ = 0.02878, squamous cell carcinoma: p = 0.6867, $R^2$ = 0.04471). A clustering by tissue type could neither be observed in Fig 2. The distances were recomputed with or without samples from both types of tissue for statistical tests. The type of cancer (adenocarcinoma or squamous cell carcinoma) did not statistically account for the variance in bacterial profile in healthy ($R^2$ = 0.0451, p = 0.09458) or cancerous tissue ($R^2$ = 0.03497, p = 0.5311). No other variable could explain variance in beta diversity. Therefore, the bacterial profile of the samples did not exhibit a global pattern. Samples from a same variable group, including type of tissue and cancer, do not have highly similar bacterial composition.

In the absence of broad bacterial profiles trends, differential abundances (OTU level) were scrutinized using DESeq2 to detect single OTUs that could be over- or underrepresented in certain tissues but might not lead to broad associations. Eleven OTUs were differently abundant (p<0.05) between the cancerous tissues of both types of cancer and thirteen from healthy tissues (Fig 3). On the other hand, no OTU was significantly more or less present when comparing the cancerous and healthy tissue within the group of patients with adenocarcinoma. Only one OTU, identified as *Phascolarctobacterium*, was 7.80 times (log2fold = -2.9626) more abundant in cancerous than healthy tissue in patients with squamous cell carcinoma. OTUs in

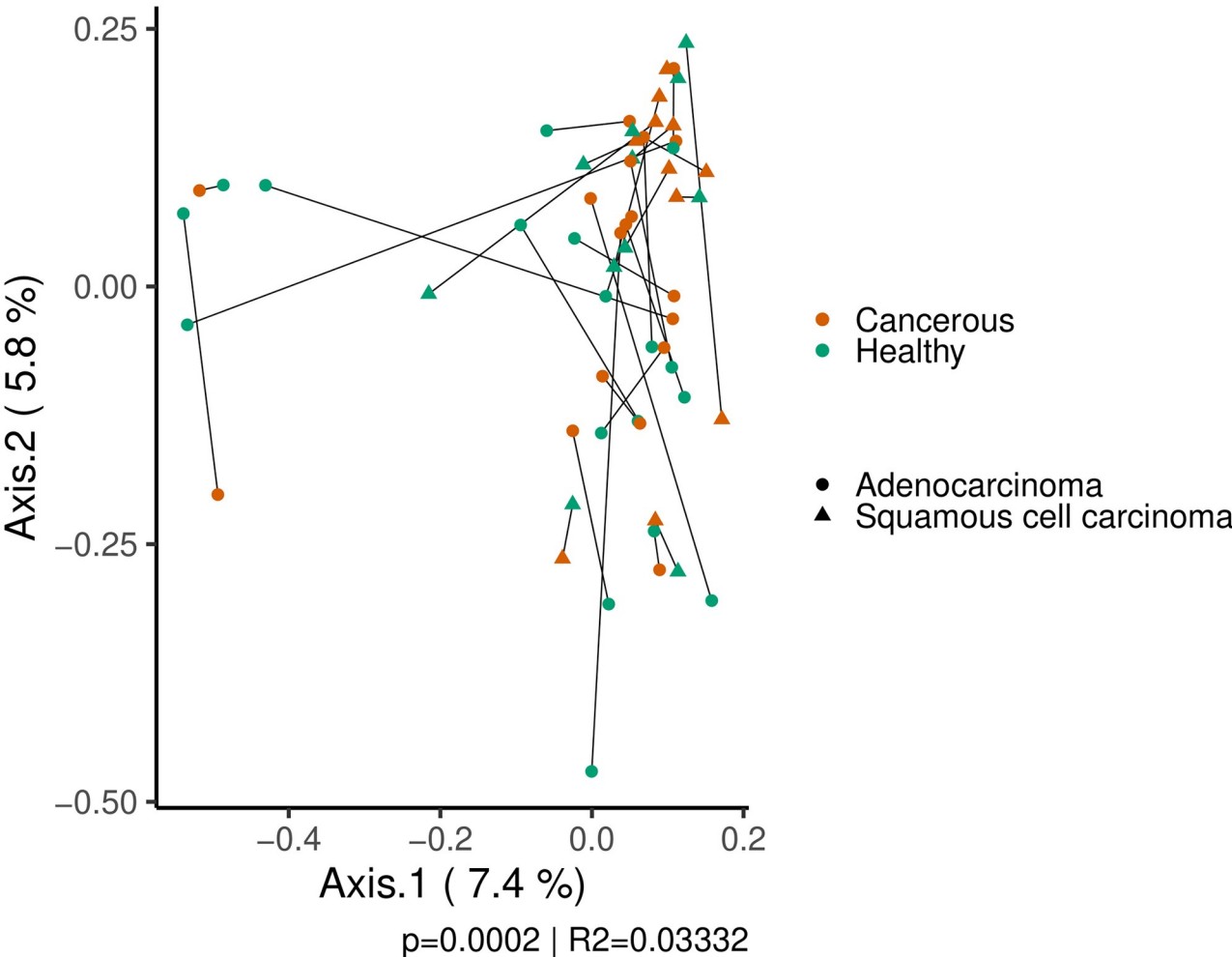

**Fig 2. Principal coordinates analysis of the weighted Bray-Curtis distances of tissue samples.** Each dot and triangles represent a tissue sample from a patient with lung adenocarcinoma or squamous cell carcinoma, respectively. Cancerous tissues are displayed in orange and healthy ones in green. The two samples from a same patient are linked by a straight line. The p-value and $R^2$ displayed are from a variance analysis (ADONIS) correlating the variation to the patients. The samples from a same patient are more similar than tissues of other patients.

tumoral and healthy tissues do not differ in a defined uniform way for each patient of a same pathology.

Pearson's correlations were computed (Table 2) to assess similarity of bacterial profiles between healthy and cancerous tissues of a same patient (intra-patient) to identify the nature of the trend observed with Bray-Curtis distances (Fig 2). Correlations vary from -1 to 1 and are less predominant when closer to 0. Pairs with a p-value over 0.05 have statistically significant correlations. Only 9 pairs out of 29 were correlated and significantly similar with coefficients ranging from 0.152 to 0.760. As previously mentioned, the presence of these correlations could not be explained by the clinical variables collected. Very few OTUs were shared between samples of a same patient compared to their total number of OTUs, even when statistically correlated (between 2 and 14 OTUs) (Table 2). However, these OTUs would represent a large proportion of the total of reads (from 20% to 65.1%) and force this correlation between the two samples. Patients 1 and 5 shared a high number of OTUs, 13 and 19 respectively, but were not significantly correlated due to important discrepancies in relative

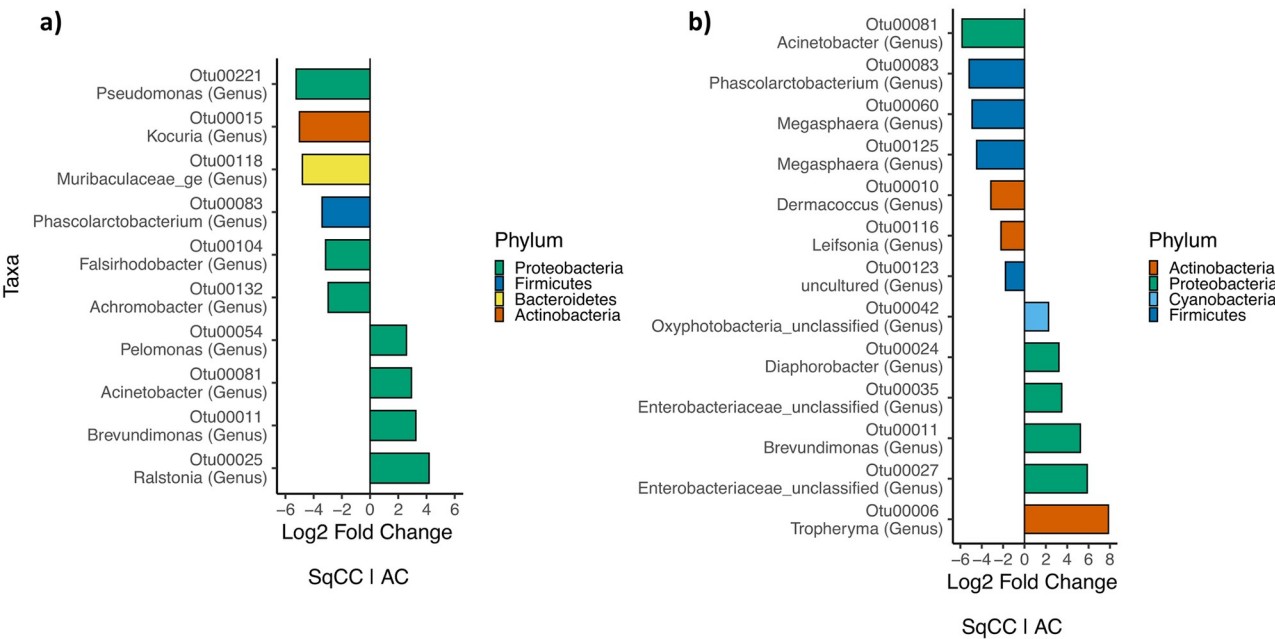

**Fig 3. Differential abundance analysis results obtained with DESeq2. a)** Differentially abundant OTUs between cancerous tissues of the two types of cancer **b)** Differentially abundant OTUs between healthy tissues of the two different types of cancer. Log2 folds changes represent the number of times the relative abundance of an OTU is doubled between cancerous tissue of adenocarcinoma (right) and squamous cell carcinoma (left). Colors represent the phylum to which the taxonomical identification of the OTUs belongs.

abundance. On the other hand, the pair of tissues from patients 15, 21, 22, and 29 did not share any OTU. The shared OTUs were from a wide range of taxonomic nature. Taxonomic identification and individual relative abundances are available in the S1 and S2 Figs. Most of the tissues from a same patient (intra-patient) present vastly different compositions. Even correlated samples have large numbers of uncommon OTUs.

Alpha diversity computations were performed to compare the intrinsic number of different bacteria and their distribution in samples based off the clinical variables. These results are available in the Supporting information file for characterization purposes but do not give particularly insightful information regarding the mechanisms and the ecological relationships involved in pulmonary microbiota (S5 and S6 Figs).

## Taxonomic analysis and core microbiota

To identify the bacteria frequently present in the lung of cancer patients, OTUs found recurrently (30% of samples) in healthy and cancerous tissues of both types of cancer were extracted. Very few OTUs were shared in samples from a same cancer type, even when healthy or cancerous were considered separately (S3 and S4 Figs). The nature of these OTUs and their relative abundance in the different samples are available in Figs 4 and 5. Comparing these two figures, many genera are revealed as abundant in both types of tissue, including *Diaphorobacter*, *Micrococcus*, *Paracoccus*, *Phascolarctobacterium*, and *Ralstonia*. However, enteric bacteria, potential pathogens, or inflammatory bacteria, such as *Escherichia-Shigella*, *Faecalibacterium*, *Pseudomonas*, unclassified *Enterobacteriaceae*, *Alloprevotella*, and *Brevundimonas*, are only recurrently present in the tumoral tissues. OTUs identified by this method also partially coincides with results from the differential abundance analysis (Fig 3), since the presence of a same OTU in multiple samples from a same group in necessary to obtain significance.

**Table 2. Intra-patient comparisons: Pearson's correlation coefficient and abundance of shared OTUs between healthy and cancerous tissue of a same patient.**

| | Pearson's correlation | | | Shared OTUs between tissue pairs | | | | |
|---|---|---|---|---|---|---|---|---|
| Patient identification number (ID) | Correlation coefficient | Significance | p-value | Number of OTUs shared | Abundance of shared OTUs in cancerous tissue (%) | Abundance of shared OTUs in healthy tissue (%) | Total number of OTUs in cancerous tissue [*] | Total number of OTUs in healthy tissue [*] |
| 25 | 0.760 | <0.05 | $2.35 \times 10^{-25}$ | 2 | 65.1 | 51.0 | 65 | 65 |
| 16 | 0.609 | | $8.55 \times 10^{-22}$ | 3 | 39.9 | 49.0 | 113 | 91 |
| 14 | 0.485 | | $3.36 \times 10^{-09}$ | 4 | 48.9 | 33.3 | 78 | 59 |
| 19 | 0.343 | | $3.51 \times 10^{-06}$ | 3 | 32.8 | 39.0 | 83 | 94 |
| 17 | 0.255 | | 0.000640 | 2 | 27.0 | 22.2 | 71 | 107 |
| 2 | 0.165 | | 0.00750 | 14 | 28.1 | 26.5 | 128 | 149 |
| 4 | 0.178 | | 0.0102 | 10 | 41.1 | 20.5 | 113 | 105 |
| 20 | 0.219 | | 0.0146 | 5 | 20.0 | 36.0 | 66 | 63 |
| 27 | 0.152 | | 0.0428 | 3 | 24.4 | 19.4 | 69 | 111 |
| 23 | 0.140 | >0.05 | 0.153 | 1 | 8.07 | 52.1 | 74 | 32 |
| 1 | 0.0740 | | 0.270 | 13 | 50.9 | 16.5 | 92 | 145 |
| 3 | 0.0621 | | 0.344 | 7 | 19.0 | 16.1 | 133 | 108 |
| 29 | -0.0890 | | 0.362 | 0 | 0.0 | 0.0 | 60 | 47 |
| 21 | -0.0798 | | 0.444 | 0 | 0.0 | 0.0 | 60 | 34 |
| 6 | -0.0300 | | 0.445 | 4 | 9.10 | 0.920 | 114 | 539 |
| 24 | -0.0512 | | 0.466 | 4 | 4.72 | 8.14 | 140 | 69 |
| 7 | -0.0456 | | 0.496 | 5 | 6.99 | 17.1 | 98 | 132 |
| 10 | -0.0460 | | 0.514 | 2 | 2.09 | 3.39 | 141 | 64 |
| 13 | -0.0461 | | 0.529 | 2 | 7.87 | 0.0414 | 125 | 66 |
| 5 | -0.0368 | | 0.543 | 19 | 63.6 | 7.11 | 144 | 151 |
| 15 | -0.0607 | | 0.559 | 0 | 0.0 | 0.0 | 48 | 47 |
| 9 | -0.0474 | | 0.560 | 1 | 3.78 | 0.0355 | 113 | 42 |
| 11 | 0.0317 | | 0.628 | 1 | 2.58 | 27.4 | 179 | 58 |
| 22 | -0.0308 | | 0.733 | 0 | 0.0 | 0.0 | 92 | 33 |
| 12 | -0.0139 | | 0.832 | 1 | 3.93 | 5.84 | 156 | 81 |
| 26 | -0.0167 | | 0.836 | 2 | 11.2 | 11.3 | 63 | 95 |
| 8 | 0.00360 | | 0.957 | 2 | 7.43 | 10.6 | 116 | 116 |
| 18 | 0.00284 | | 0.974 | 3 | 6.97 | 11.2 | 79 | 59 |
| 28 | 0.00138 | | 0.988 | 3 | 9.16 | 35.6 | 57 | 58 |

[*]The number of OTUs were not obtained through rarefication and do not account for different in sequencing depth.

## Discussion

This study leverages a rigorous sampling protocol previously reported by our team to further document the composition of pulmonary microbiota in lung cancer and to minimize the impact of contaminations. Pairs of cancerous and healthy pulmonary tissue from 29 non-small cell lung cancer patients with their corresponding methodological controls were analyzed.

### Accounting for contaminants

As described in the previously published method article [22], methodological controls, accounting for every step of the experimental protocol, from sampling to sequencing, were

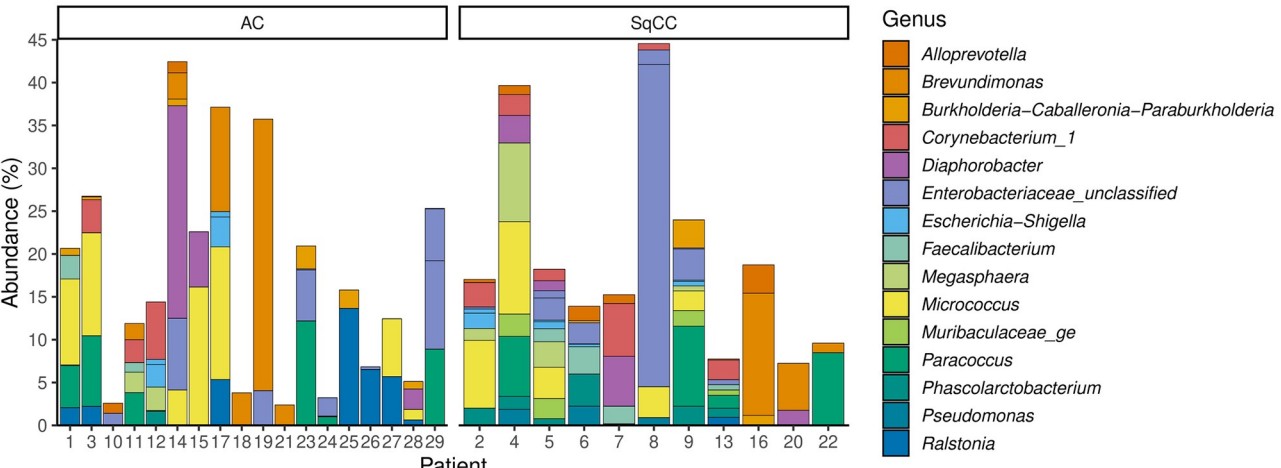

**Fig 4. Taxonomic identification and relative abundance of OTUs found in 30% of cancerous samples in either adenocarcinoma, squamous cell carcinoma or both type of tumor.** Each section of bar plots represents one OTU and each color a different genus. This figure matches the OTUs identified in the S3 Fig.

added for each patient. They are representative of the bacterial contaminants that could have been introduced in the samples throughout the protocol. The bacterial signature of control, as measure by the weighted and unweighted Bray-Curtis beta diversity metric, was significantly different (p = 0.0002) to the corresponding samples (Fig 1). Therefore, the tissue samples and the controls share very few OTUs in potentially different proportions and the influence of contaminants on the bacterial composition detected for tissues seems negligible. The removal of the contaminating OTU was still performed as described by Dumont-Leblond *et al*. [22,24]. The results presented here are a reliable description of the pulmonary microbiota.

## Influence of clinical variables

**Patient correlation.** Tissues from a same patient correlated more together (intra-patient) than they do with others (inter-patient) as a whole from weighted Bray-Curtis distances

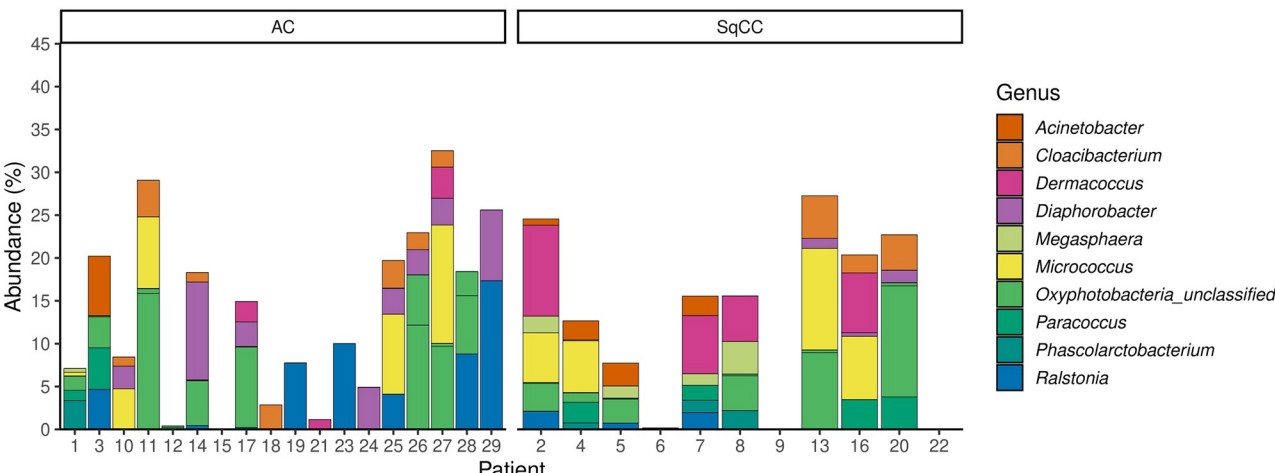

**Fig 5. Taxonomic identification and relative abundance of OTUs found in 30% of healthy tissue in either adenocarcinoma or squamous cell carcinoma afflicted patients.** Each section of bar plots represents one OTU and each color a different genus. This figure matches the OTUs identified in the S4 Fig.

(Fig 2). However, there seems to be a lot of variations as to how much cancer and healthy tissues from a same patient are similar. In fact, the Pearson's correlations that were computed on the relative abundance of OTUs show that only 9 of the 29 patients (31%) of tumor-healthy pairs were significantly correlate (Table 2) considering the presence of a same OTU and its relative abundance. These samples were most probably the origin of the broader correlation by patients observed previously. The difference in clustering observed in Fig 2 and the low amount of variance explained by the patient variable ($R^2 = 0.03332$) result from this majority of tissue pairs that do not have strong correlations. It is also explained by the low number of OTUs shared by tissue pairs (from 2 to 14 in significantly correlated pairs) compared to their total number of OTUs (Table 2). They, however, may represent a majority of the total relative abundance of the samples (from 19.4% to 65.1%). Therefore, every cancerous and tissue sample pairs have a large proportion of OTUs they do not share, even in highly correlated ones. Even in relative proximity, tissues from a same patient do not harbor identical bacterial communities. Urbaniak et al. also found large discrepancies between healthy and cancerous breast tissues from a same patient who underwent lumpectomies or mastectomies [40]. Similarly, Nejman *et al.* observed partial similarity between cancerous and adjacent tissue of a same cancer (colon, breast, lung, ovary) [4].

**Type of tissue and cancer.**   Tumoral tissues and healthy tissues from different patients (inter-patient) do not cluster into two distinct groups (Fig 2). The bacterial profiles of the two types of tissue from different patients are not similar in adenocarcinoma (p = 0.42352, $R^2 = 0.02878$) or squamous cell carcinoma (p = 0.6867, $R^2 = 0.04471$). It implies that there is not a characteristic microbiota composition for cancerous or healthy tissue in lung cancer. This was not previously reported as lung cancers were usually considered as a whole, without discrimination for the subtypes. The lack of relationship between the healthy tissue (inter-patient) may also indicate that patients have a distinct bacterial signature or baseline as is observed in gut microbiota [41]. These differences may make it difficult to uncover the influence of variables on bacterial composition of cancer microbiota as a similar factor applied on communities of different composition may not lead to the same alterations. In an attempt to counterbalance this patient specific effect, both healthy and cancerous tissue were sampled to observe non-pathologic microbiota and create a baseline for each individual. We then attempted to explain the variation in the dataset by analyzing the differences between the pairs and identify OTUs that were significantly more or less present between cancerous and healthy tissue for each of the two types of cancer. No OTU could be identified as differentially abundance across the cohort of patients with adenocarcinoma (18 patients) and only one with squamous cell carcinoma (11 patients), classified as *Phascolarctobacterium*. This genus includes species such as *Phascolarctobacterium faecium* and *Phascolarctobacterium succinatutens* that are known colonizers of the intestinal tract [42,43]. The abundance of individual OTUs between cancerous and healthy tissue of patients of a same pulmonary cancer (inter-patient) do not vary in a characteristic way. The lack of strong correlation to begin with for every pair of tissue (intra-patient) may explain why relative abundance analyses proved unsuccessful, as relative abundance metrics can only be computed when a same OTU is present in both samples.

Considering these results and this study design, we are unable to identify what may cause those major variations on bacterial profiles within the type of tissue (cancer vs. healthy). We hypothesize that two mechanisms, that may be simultaneous, are possible:

1. The tumoral tissue could be colonized by a specific microbiota due to its different matrix or human cellular activity, which would explain the large disparity between its bacterial composition and that of the healthy tissue is most cases. The intrinsic characteristics of the tumor, e.g., the histologic pattern (lepidic, acinar, papillary, etc.) and genetic background,

may have an additive influence on the tumoral ecosystem. Pairs of tissues that are more similar would demonstrate a lower level of specific colonization that may or may not develop as the pathology progresses. The tumoral environment could also have an influence on the bacterial profiles of healthy tissue, but not significantly enough to create ecological niches that would be occupied by the same bacteria.

2. The tumor could have no significant effect on the bacterial composition of the lung. The presence of a transitory microbiota that has little growth and residency due to low nutrient availability could be very mildly affected by the tumorous environment. This could explain the frequent differences in microbiota composition by patients and type of tissue, since the composition of the microbiota would become dependent on the local impaction of bacteria and clearance by the host, phenomenon knows to occur in the lungs [44].

Following the variation in time of the microbiota and its potential shift when a lung cancer is developed may be able to elucidate part of these theories. Such longitudinal studies of human candidates may not be possible as it would require repeated removal of lung tissue, but murine models could help get better insight into the community-shaping forces at play.

## Taxonomic analysis and core microbiota

In the absence of broad bacterial associations between the type of cancer with Bray-Curtis distances (Fig 2), differential abundance analyses were performed to identify individual OTUs that were over- or underrepresented in the same type of tissue from the two different cancers (Fig 3). Ten differently abundant OTUs (6 more in SqCC and 4 in AC) in cancerous tissues and 13 in healthy tissues were reported. It is worth nothing that tissues from different types of cancer also come from different patients and this effect cannot be accounted for. Considering the high inter-individual variation previously observed, caution is advised when interpreting these results.

The presence of a bacterial genus in at least 80% of samples usually defines the term "core microbiota". In the absence of any genus present in that many samples, 30% was used as a baseline to characterize frequently present genera. The lack of similarity between samples may be explained by the different mechanisms of colonization and elimination observed in the pulmonary microbiota compared to other environments, such as the gut microbiota [45].

Even with this low threshold, very few OTUs were recurrently identified and represented a fairly low average relative abundance, below 21% (S3 and S4 Figs). The taxonomic classification of these OTUs were extracted and displayed in Figs 4 and 5 by type of tissue and cancer. The genera *Diaphorobacter*, *Micrococcus*, *Paracoccus*, *Phascolarctobacterium*, and *Ralstonia* were shared by both types of tissue. Enteric or potentially pathogenic genera, such as *Escherichia-Shigella*, *Faecalibacterium*, *Pseudomonas*, unclassified *Enterobacteriaceae*, *Alloprevotella*, and *Brevundimonas*, were only present in cancerous tissue. Such trend had never been observed in previous literature, to our knowledge. The chances of specific impaction of these genera only on cancerous tissue seems unlikely. The tumoral environment must allow the replication and survival of those strains for them to be frequently detected. It is impossible to tell if these bacteria had any responsibility in carcinogenesis or if their presence results from the modification of the pulmonary environment by the apparition of cancer. As reported in gut microbiota, it is not impossible for these bacteria to be interfering with anti-cancerous treatments [7,20].

While performing bronchoalveolar lavage on healthy smokers, Erb-Downward *et al.* found in 75% of their samples the genera *Pseudomonas*, *Streptococcus*, *Prevotella* and *Fusobacterium*. *Haemophilus*, *Veillonella* and *Porphyromonas* were also found in 50% of the samples [1]. No

genera were observed as frequently present. With the exception of *Prevotella* and *Fusobacterium*, these genera were detected throughout the dataset, but not recurrently in the healthy tissues. These disparities could be attributed to the different sampling and sequencing methods, to lack of obvious concerns for contaminants and the absence of diagnosed cancer in the patient they sampled.

While analyzing the intratumoral lung microbiota with whole-tissue DNA extraction, Yu *et al.* also frequently detected the following genera: *Burkholderia*, *Corynebacterium*, *Pseudomonas*, and *Rasltonia*. Yet, the present study reports a greater number of different genera, including enteric bacteria. Their method was more alike to the one used in this article than Erb-Downward *et al.* The slight difference, such as the use of different bioinformatics tools (potentially 16s rRNA gene database), might still have explain part of the dissimilarities. They also do not seem to differentiate the different subtypes of lung cancer.

A recent study by Nejman *et al.* reported the characterization of the intratumoral microbiota of 245 cancerous lung samples [4]. Great precautions were taken to limit the introduction of contaminants during samples and data treatment, but little is known on the tissue sampling process and most of the patient's clinical data is undisclosed. They identified the frequently found taxa at the family level. They detected every bacterial family related to the genera identified in Fig 4 (10; *Provetellaceae*, *Pseudomonadaceae* [*Burkholderiaceae*], *Corynebacteriaceae*, *Comamonadaceae*, *Veillonellaceae*, *Micrococcaceae*, *Rhodobacteraceae*, *Pseudomonadaceae*), with the exception of *Caulobacteraceae* (*Brevundimonas*), *Muribaculaceae*, *Acidaminicoccoceaceae* (*Phascolartobacterium*), and *Rasltoniaceae* (*Ralstonia*). These discrepantly could be associated with the vastly dissimilar methodologies used (sequencing method; V3-V4 vs. R5 multiplex, 16S rRNA gene database; Greengenes vs. SILVA, custom bioinformatics pipelines) or the lack of subdivision in types of cancer.

## Limitations and perspectives

Relatively low number of biological replicates used in this study might have reduced our ability to detect significant patterns in the microbial diversity. However, it might not be that clinical variables do not have an influence on the bacterial composition, but that their effects are additive to other determinants of the lung microenvironment and do not have the same community shaping effects from one individual to another. The host-microbiota relationship may be too complex and specific to allow for the detection of large patterns with tools currently available.

In fact, discriminating the effect of those multiples contributing factors and identifying an optimal microbial consortium for an individual are currently two of the main challenges in term human microbiota study. Moving forward, scientists may benefit from looking into machine-learning algorithms [46] and artificial intelligence (AI) to allow a more in-depth analysis of the large amount of microbiota data currently available or easily obtainable with new generation sequencing techniques. The quality of data and methodology used to collect them should be taken in considerations as models can only be as accurate as the data their stem from, hence our previous push in lung microbiota methodology standardization.

On the other hand, NGS methods currently available should be perceived as prospective, as their high data production power and complexity (multiple steps) makes them susceptible to numerous methodological biases. The absence of absolute quantification may also be limiting. Therefore, the combination of NGS and quantitative methods such as qPCR may help provide a more complete overview of the biological community and prevent over analysis of highly processed NGS data by providing reference points and counter verification. In this study we

present prospective data for the development of more focused methodologies and elucidation of the dynamic forces shaping the lung microbiota.

As observed in gut microbiota studies, same bacterial genera may contain different genes and functions [47]. Our description of the microbial flora through selective amplification and NGS might not have achieved the level of resolution required to pick up underlying trends. Omics techniques (metatranscriptomics and metabolomic) applied to the lung microbiota may provide this additional resolution and seems to be the next step in unveiling the microbiota role in the lung and their influence on pulmonary health.

On the other end, the microbial characterization of healthy tissue achieved here may not be representative of bacterial communities found in patients without cancer. The possibility that the cancerous microenvironment or the systemic influence of cancer may have a significant impact on the adjacent tissues cannot be rejected. Sampling healthy tissue from patients who undergo resection for benign pulmonary disease may provide a more comprehensive representation of the healthy lung microbiota.

This study presents the characterization of the lung microbiota from cancerous and healthy tissues of twenty-nine non-small-cell cancer patients using a rigorous 16S rRNA sequencing methodology. The cancerous and healthy tissue samples from a same patient (intra-patient) were more similar than the ones from a different individual (inter-patient). The composition of bacterial communities could not be correlated to the variables we collected, including type of cancer and type of tissue. The recurrent presence of enteric and proinflammatory bacteria characterized the cancerous lung tissue. Further work is needed to understand the importance of lung microbiota on the human health and its influence on outcomes in cases of pulmonary cancer. This work is one of the early in-depth characterizations of whole-tissue lung microbiota in cases of pulmonary adenocarcinoma or squamous cell carcinoma. It establishes strong bases and recommendations for the pursuit of this field of study.

## Supporting information

**S1 Fig. Relative abundance and taxonomic identification of OTUs shared between the two tissue samples from a same patient correlated by Pearson's test (p-value<0.05).** Each section of the bars represents one OTU. Each bacterial genus is identified with a different color. The type of tissue, cancerous or healthy, is specified by a full or dotted outline, respectively. (DOCX)

**S2 Fig. Relative abundance and taxonomic identification of OTUs shared between the two tissues samples from a same patient not correlated by Pearson's test (p-value>0.05).** Each section of the bars represents one OTU. Each bacterial genus is identified with a different color. The type of tissue, cancerous or healthy, is specified by a full or dotted outline, respectively. (DOCX)

**S3 Fig. Number of OTUs shared by 30% of all cancerous tissues for each type of cancer with relative abundances higher than 0.001%.** The numbers in parentheses represent the sum of the average relative abundance of the OTUs in the samples. The non-core score is the number of OTUs with relative abundances higher than 0.001% that are not present in at least 30% of the sample one of the categories. (DOCX)

**S4 Fig. Number of OTUs shared by 30% of all healthy tissues for each type of cancer with relative abundances higher than 0.001%.** No OTUs are shared between the two types of

cancer. The numbers in parentheses represent the sum of the average relative abundance of the OTUs in the samples. The non-core score is the number of OTUs with relative abundances higher than 0.001% that are not present in at least 30% of the sample one of the categories.
(DOCX)

**S5 Fig. Alpha diversity of tissue samples by type of cancer. A**. Shannon's alpha diversity index. **B**. Pielou's evenness. Doubled-sided t-tests paired by patients were performed. The boxes display the data range, quartiles and median. The mean is displayed as white lozenges.
(DOCX)

**S6 Fig. Comparison of Shannon's diversity index of tissue samples by type of cancer.** Doubled-sided non-paired t-tests were performed. The boxes display the data range, quartiles and median. The mean is displayed as white lozenges.
(DOCX)

**S1 Table. Accession number of the raw sequencing data in the project PRJNA680529 of the NCBI sequence read archive.**
(DOCX)

# Acknowledgments

For their involvement in patient recruitment and sampling supervision, we thank Marie-Christine Allard and Marie-Ève Côté from the QRHN biobank—site IUCPQ. We also acknowledge the contribution of surgeons and pathologist for their cooperation in the sampling campaign, as well as the graduate students from the Duchaine's laboratory for their support with instruments management.

# Author Contributions

**Conceptualization:** Nathan Dumont-Leblond, Marc Veillette, Christine Racine, Philippe Joubert, Caroline Duchaine.

**Data curation:** Nathan Dumont-Leblond, Christine Racine.

**Formal analysis:** Nathan Dumont-Leblond.

**Funding acquisition:** Nathan Dumont-Leblond, Marc Veillette, Philippe Joubert, Caroline Duchaine.

**Investigation:** Nathan Dumont-Leblond.

**Methodology:** Nathan Dumont-Leblond.

**Project administration:** Nathan Dumont-Leblond, Christine Racine.

**Resources:** Christine Racine.

**Software:** Nathan Dumont-Leblond.

**Supervision:** Marc Veillette, Philippe Joubert, Caroline Duchaine.

**Visualization:** Nathan Dumont-Leblond.

**Writing – original draft:** Nathan Dumont-Leblond.

**Writing – review & editing:** Marc Veillette, Christine Racine, Philippe Joubert, Caroline Duchaine.

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
