## [Decision Letter · Decision Letter 0]

29 Jan 2021

PONE-D-20-39437

Non-small Cell Lung Cancer Microbiota Characterization: Prevalence of Enteric and Potentially Pathogen Bacteria in Cancer Tissues

PLOS ONE

Dear Dr. Duchaine,

Thank you for submitting your manuscript to PLOS ONE. After careful consideration, we feel that it has merit but does not fully meet PLOS ONE’s publication criteria as it currently stands. Therefore, we invite you to submit a revised version of the manuscript that addresses the points raised during the review process.

Both reviewers were enthusiastic about this work and thought it represented an important knowledge gain.  Reviewers suggested minor revisions to add clarity to the methods, as well as suggestions to help focus the Discussion section.

We look forward to receiving your revised manuscript.

Kind regards,

Suzanne L. Ishaq, PhD

Academic Editor

PLOS ONE

Journal Requirements:

Reviewers' comments:

Reviewer's Responses to Questions

**Comments to the Author**

1. Is the manuscript technically sound, and do the data support the conclusions?

Reviewer #1: Yes

Reviewer #2: Partly

2. Has the statistical analysis been performed appropriately and rigorously? 

Reviewer #1: Yes

Reviewer #2: Yes

3. Have the authors made all data underlying the findings in their manuscript fully available?

Reviewer #1: Yes

Reviewer #2: No

4. Is the manuscript presented in an intelligible fashion and written in standard English?

Reviewer #1: Yes

Reviewer #2: Yes

5. Review Comments to the Author

Reviewer #1: The paper by Dumont-Leblond et al provides new insights into the relationship between commensal bacterial communities (microbioma) in the lung and the development of lung cancer. This is a very interesting field that is just emerging and thus these results significantly advance our knowledge. Moreover, it is meritorious the novel methodology employed to ensure the quality of the analyses and avoid contaminations of the high respiratory track, which might produce cofounding results when analyzing lung microbial communities.

I only have some minor comments on the manuscript.

1- The paper is well written and deals with an interesting and timely topic such as the relation of lung microbiota and pulmonary cancer. The work is well organized and comprehensively described but the discussion is too long and it is not easy to keep focused the whole time. I would recommend focusing the discussion on the most relevant findings.

2- A main limitation of the study is the comparison of microbiota in biopsies from the same lung lobe which might difficult the comparisons between “healthy” and cancer tissue. I know it is challenging to take samples from the non affected lung but this should be remarked in the paper. Would it be possible to get some information of the microbioma from healthy lung from published data bases in order to compare? I realize the methods for the results obtained in those databases might be different but at least there should be a cohort to compare.

3- I am not really sure about the meaning of another MS draft included in the resvison. I guess it is another paper under evaluation focused on the validation of the methodology employed for the current analyses. If so, this should be clarified and the MS referenced in the text. I might have missed it but did not find such a reference.

4- An idea to shorten the discussion might be including references to recent reviews analysing the role of lung microbiota in lung cancer. A suggestion, Ramirez-Labrada et al Trends Cancer 2020.

Reviewer #2: Overall, this is an interesting study that appears to be rigorously performed. Two concerns limit my enthusiasm. 1) Reproducibility: The authors appear to have split the work performed here into two papers, spinning the methodological details off into a separate submission. While they did include this methods paper, the removal of this data reduces the reproducibility of this particular paper. Secondarily, while the authors have uploaded their raw sequencing data, the remaining data is included here only as “Underlying data”. The authors should upload the OTU tables used, and the analysis code to a publicly accessible repository such as GitHub or include it here if it will be available with the article. 2) Small sample size. The rigor and reproducibility of this work is therefore reduced, especially in light of studies with larger sample sizes.

Minor edits:

Line 213-15: These details need to be described in text (with a citation).

Line 232: Typo. ‘arbor’ is not the right word, likely harbor was intended.

Line 239-40: This paragraph is unclear, due to both language and lack of detail. Please clarify the text.

Line 272 the phrase ‘is a majority of cases’ does not make grammatical sense

Line 276 Similar, ‘It is not discarded’ does not make sense

Line 434: The authors should specify what methods were used to adjudicate contaminates (was the DECONTAM package used ? If not then please reanalyze using this package).

6. PLOS authors have the option to publish the peer review history of their article (what does this mean?). If published, this will include your full peer review and any attached files.

Reviewer #1: No

Reviewer #2: No

---

## [Author Response · Author response to Decision Letter 0]

19 Feb 2021

Authors are very thankful for the thorough revision of our manuscript and fruitful comments. Here is a point-by-point response to reviewers’ comments and highlights of major changes.

The reviewers seemed uneasy with the fact that this work refers heavily on the provided method manuscript for the description of its methodology. We agree that the format in which it was provided to them was not ideal, since the article was still in the publication process, and that it may have hindered the clarity of this work. However, it has now been published in Communications Biology, as of February 5 ( https://www.nature.com/articles/s42003-021-01690-5#Sec21 ). We hope it will be both a sign of the rigor of our work and that the new format will facilitate their understanding. As seen below, we added some methodological precision when requested, but we believe going in depth into methodological details would only make the manuscript heavier since these are already available in an open access journal. The sections of the article have been rearranged to fit the journal requirements, which now places the Materials and Methods section prior to the Results sections. We hope this will also make the article easier to understand when read from top to bottom.

Reviewer 1

1- The paper is well written and deals with an interesting and timely topic such as the relation of lung microbiota and pulmonary cancer. The work is well organized and comprehensively described but the discussion is too long and it is not easy to keep focused the whole time. I would recommend focusing the discussion on the most relevant findings.

We thank you and are glad that you appreciated this manuscript. We understand your concerns regarding length. This manuscript remains highly technical in its analysis and the different metrics used. We removed a small section of the discussion. However, we believe that the overall length (≈ 5 pages, 1.5 spacing) is not out of the ordinary. We also did our best to divide it clearly in 4 sections and only addressing the 3 major comparisons (intra vs. inter patients, healthy vs. tumoral tissue, adenocarcinoma vs. squamous cell carcinoma). Reinterring the reliability of our data (‘Accounting for contaminants’), comparing our results to others, and outlining the limitation and perspective of this study also seemed mandatory. The other reviewer did not express a similar concern and we wish to keep the manuscript in this now reviewed format.

2- A main limitation of the study is the comparison of microbiota in biopsies from the same lung lobe which might difficult the comparisons between “healthy” and cancer tissue. I know it is challenging to take samples from the non affected lung but this should be remarked in the paper. Would it be possible to get some information of the microbiome from healthy lung from published data bases in order to compare? I realize the methods for the results obtained in those databases might be different but at least there should be a cohort to compare.

We have acknowledged the limitations in the second to last paragraph of the manuscript and have characterized extensively the relation between tumoral and adjacent tissues throughout the article:

Line 539:

“On the other end, the microbial characterization of healthy tissue achieved here may not be representative of bacterial communities found in patients without cancer. The possibility that the cancerous microenvironment or the systemic influence of cancer may have a significant impact on the adjacent tissues cannot be rejected. Sampling healthy tissue from patients who undergo resection for benign pulmonary disease may provide a more comprehensive representation of the healthy lung microbiota.”

To our knowledge, published data of whole-tissue healthy lung microbiota is not available. Has mentioned, access to such tissue is very limited, hence our attempt at using adjacent healthy tissue. Even if such data was available, the divergence of methodology would make a comparative analysis very weak. We do not proclaim that the microbiota detected is representative of the one found in individuals that do not have cancer. The comparisons between tumoral and adjacent healthy tissue remains of interest to understand to possible preferential bacterial colonization of cancerous tissue.

3- I am not really sure about the meaning of another MS draft included in the revision. I guess it is another paper under evaluation focused on the validation of the methodology employed for the current analyses. If so, this should be clarified, and the MS referenced in the text. I might have missed it but did not find such a reference.

As stated above, we recognize that the format in which the method article was provided to you was not ideal. It has now been published. It was already cited numerous times throughout the manuscript. The reference has been updated (line 755):

22. Dumont-Leblond N, Veillette M, Racine C, Joubert P, Duchaine C. Development of a robust protocol for the characterization of the pulmonary microbiota. Commun Biol. 2021;4: 164. doi:10.1038/s42003-021-01690-5

4- An idea to shorten the discussion might be including references to recent reviews analysing the role of lung microbiota in lung cancer. A suggestion, Ramirez-Labrada et al Trends Cancer 2020.

Thank you for the suggestion. We do not feel that this article can replace any part of the discussion. However, we added this reference to the introduction. 

Reviewer 2

1- Reproducibility: The authors appear to have split the work performed here into two papers, spinning the methodological details off into a separate submission. While they did include this method paper, the removal of this data reduces the reproducibility of this particular paper. 

The methodological paper is very extensive. We believed that the complexity of the methodology, which may not be apparent at first sight, required its dedicated publication. Combining the information of the two papers would have resulted in a way too long article. Reviewer 1 also requested that we shorten this manuscript. We do not believe that it reduces the reproducibility of this paper in any way, since the details of methodology remains available in an open access journal (Communications Biology). Referring to protocols from other publications is somewhat common practices. We still included enough methodological details here to ensure proper understanding without making the manuscript too heavy. 

2- Secondarily, while the authors have uploaded their raw sequencing data, the remaining data is included here only as “Underlying data”. The authors should upload the OTU tables used, and the analysis code to a publicly accessible repository such as GitHub or include it here if it will be available with the article. 

The OTU and taxonomic tables can already be found in the Underlying data file (last tabs). We agree that it was not mentioned clearly enough in the Data availability subsection. The last sentence of this subsection has been modified:

Underlying data of the figures of the article, as well as the OTU and taxonomic tables, are made available in the Supporting data file. (Line 574)

The analysis code is already referred to in Material and methods section. 

Line 156: A custom contaminants removal method[23] was applied before or after diversity and differential abundance analyses were performed in RStudio [30]

Reference 23 : 

Dumont-Leblond N. Lung_Microbiota_Contaminants_Management. 2020 [cited 5 Oct 2020]. Available: https://github.com/NDumont-Leblond/Lung_Microbiota_Contaminants_Management/blob/master/Script.R

3- Small sample size. The rigor and reproducibility of this work is therefore reduced, especially in light of studies with larger sample sizes.

We are aware of this limitation of our study, but do not believe it renders it invalid. As per the nature of current next generation sequencing methods and their biases, this study remains exploratory and we acknowledge its limitations in the Limitations and perspectives subsection. We are unable to increase to sample size due to the ongoing pandemic and the clinical research limitations. 

Minor edits:

Line 213-15: These details need to be described in text (with a citation).

We agree that this part requires clarifications. By moving the Materials and methods before the results section, we believe the reader will have a better understanding the method (Contaminants management and Sequences processing subsections), before reading the discussion. The sentence was modified:

The removal of the contaminating OTU was still performed as described by Dumont-Leblond et al [22,24]. 

Line 232: Typo. ‘arbor’ is not the right word, likely harbor was intended.

Thank you for bringing this typological mistake to our attention. It has been fixed.

Line 239-40: This paragraph is unclear, due to both language and lack of detail. Please clarify the text.

We remove this paragraph since it did not add any pertinent information. It was only a short preamble to this section and, as you pointed out, it was more confusing than helping. 

Line 272 the phrase ‘is a majority of cases’ does not make grammatical sense

It has been replaced with “in most cases”.

Line 276 Similar, ‘It is not discarded’ does not make sense

The expression was removed. The rest of the sentence remains untouched.

Line 434: The authors should specify what methods were used to adjudicate contaminants (was the DECONTAM package used ? If not then please reanalyze using this package)

We did not use the DECONTAM package, but the method described in the joined article. The package was also suggested during its publication process (see below). Furthermore, considering the extremely low number of shared OTUs between samples and controls, very few of them ended up being removed (Fig 1). This step would probably not have been necessary in these circumstances but it was performed anyway to ensure all the precautions were taken.

Here is the justification that was given and that can be found in the peer review file (https://www.nature.com/articles/s42003-021-01690-5#Sec21):

‘We looked into the decontam package the reviewer is suggesting. It incorporates two types of contaminants removal, by frequency of sequence or prevalence throughout the dataset. As expressed by Davis et al., frequency-base contaminant identification is not recommended for extremely low-biomass samples. It is therefore not applicable in lung microbiota. Furthermore, the quantities of DNA measured after amplification that are required by this method cannot be precisely obtained from our samples as a large quantity of human DNA from lung cells is also present and throws off the calculations. 

The second method (prevalence-based) does not present these limitations. The function the authors presented is based on a chi-squared or fisher test performed on a 2x2 absence-presence and either control/sample table. For each OTU a table is computed to compare the group of samples and controls. If the OTU is more statistically present in the sample group, it is considered as not contaminants. This method has shortcomings as it does not account for proportion of samples. Therefore, an OTU present in only one copy (that might have been wrongly classified) in a control yields the same weight as reads of this same OTU accounting for 20% (e.i. 4000 reads) of a sample reads. Furthermore, lung microbiota characterization data that we have collected since the development of this methodology do not show strong correlation in bacterial composition of samples between patient and the absence of a define “core microbiota” as seen in gut microbiota [present manuscript]. The absence of recurring OTUs may undermine our ability to obtain a statistical difference between samples and control which would underestimate the diversity present in samples. The nature of the method is also not compatible with our study design. As acknowledge by the authors, the contaminant removal method is more robust as the number of samples and control increase. Our study design incorporates a single control for each pair of samples from each patient. These controls account for steps that are not shared by every sample we analyzed, as least not necessarily in the same batch, and cannot be pooled to allow for the global analysis that is recommended. Performing the analysis on pairs does not seem appropriate as the sensitivity of this method is limited when few samples are compared due to the nature of the statistical test used. 

To our knowledge, no other method could be compared in the case of lung microbiota and that is compatible with the study design presented in this article. On the other hand, as identified by the Pearson’s correlation we performed on pairs of control and sample, the sample extracted using the Blood DNA extraction kit share very little OTUs with their controls, reducing the need for a highly robust bioinformatics contaminant removal protocol. We still consider the method we present to be appropriate, but we added a sentence on our lack of deep characterization on the robustness of our method. We also referenced the study suggested by the reviewer mentioning the concerns we expressed here. 

Lines 460-463 : Some research groups tried to use a neutral community model (Venkataraman et al. 2015), additional qPCR data (Lazarevic et al. 2016), amplicon DNA yield, or prevalence algorithms (Davis et al. 2018) to assess the influence of methodological contaminants. 

Lines 481-489 : We acknowledge that our contaminant management method does not have the in-depth validation of other methods, such as described by Davis et al. with the decontam package (Davis et al. 2018). However, it does not share its limitations regarding the lack of consideration for OTU abundance and need of high number of controls to ensure sensitivity while using prevalence-based detection. Further research focused on the development of statistical methods to detect contaminant OTUs in the cases of lung microbiota is needed. This work is to be a starting point toward methodological standardization and its modular nature makes the bioinformatic contaminant management method we proposed interchangeable once a more robust one is uncovered. ‘

Additional changes performed

1- Multiple small modifications were done to comply with the publishing requirements:

a. All the titles were switched to sentence case. 

b. The font size of the titles was adjusted. They are no longer underlined.

c. The authors section was reformatted.

d. The sections were rearranged (Materials and method now comes second)

e. The acronym “Fig” now replaces the complete appellation when referring to figures in the manuscript. The format “S# Fig.” was implemented in the supporting information file.

f. The reference style was switched to the “Plos One” format.

g. The funding sources were removed from the Acknowledgements, as well as the Competing interests section.

2- Occasional grammatical or topographical mistakes were corrected (see tracked manuscript).

---

## [Decision Letter · Decision Letter 1]

26 Mar 2021

Non-small Cell Lung Cancer Microbiota Characterization: Prevalence of Enteric and Potentially Pathogen Bacteria in Cancer Tissues

PONE-D-20-39437R1

Dear Dr. Duchaine,

We’re pleased to inform you that your manuscript has been judged scientifically suitable for publication and will be formally accepted for publication once it meets all outstanding technical requirements.

Kind regards,

Suzanne L. Ishaq, PhD

Academic Editor

PLOS ONE

Additional Editor Comments (optional):

Reviewers' comments:

Reviewer's Responses to Questions

**Comments to the Author**

1. If the authors have adequately addressed your comments raised in a previous round of review and you feel that this manuscript is now acceptable for publication, you may indicate that here to bypass the “Comments to the Author” section, enter your conflict of interest statement in the “Confidential to Editor” section, and submit your "Accept" recommendation.

Reviewer #1: All comments have been addressed

Reviewer #2: All comments have been addressed

2. Is the manuscript technically sound, and do the data support the conclusions?

Reviewer #1: Yes

Reviewer #2: Yes

3. Has the statistical analysis been performed appropriately and rigorously? 

Reviewer #1: Yes

Reviewer #2: Yes

4. Have the authors made all data underlying the findings in their manuscript fully available?

Reviewer #1: Yes

Reviewer #2: Yes

5. Is the manuscript presented in an intelligible fashion and written in standard English?

Reviewer #1: Yes

Reviewer #2: Yes

6. Review Comments to the Author

Reviewer #1: (No Response)

Reviewer #2: The authors have mostly addressed my concerns, seeing as the methods paper is already published that cannot be satisfactorily addressed but it is what it is.

7. PLOS authors have the option to publish the peer review history of their article (what does this mean?). If published, this will include your full peer review and any attached files.

Reviewer #1: No

Reviewer #2: No

---

## [Editor Report · Acceptance letter]

15 Apr 2021

PONE-D-20-39437R1 

Non-small cell lung cancer microbiota characterization: prevalence of enteric and potentially pathogenic bacteria in cancer tissues 

Dear Dr. Duchaine:

I'm pleased to inform you that your manuscript has been deemed suitable for publication in PLOS ONE. Congratulations! Your manuscript is now with our production department. 

Kind regards, 

on behalf of

Dr. Suzanne L. Ishaq 

Academic Editor

PLOS ONE